# Testicular somatic cell-like cells derived from embryonic stem cells induce differentiation of epiblasts into germ cells

Holly Rore [1], Nicholas Owen [2], Raul Eduardo Piña-Aguilar [1], Kevin Docherty[1] & Ryohei Sekido [1,2]✉

Regeneration of the testis from pluripotent stem cells is a real challenge, reflecting the complexity of the interaction of germ cells and somatic cells. Here we report the generation of testicular somatic cell-like cells (TesLCs) including Sertoli cell-like cells (SCLCs) from mouse embryonic stem cells (ESCs) in xeno-free culture. We find that Nr5a1/SF1 is critical for interaction between SCLCs and PGCLCs. Intriguingly, co-culture of TesLCs with epiblast-like cells (EpiLCs), rather than PGCLCs, results in self-organised aggregates, or testicular organoids. In the organoid, EpiLCs differentiate into PGCLCs or gonocyte-like cells that are enclosed within a seminiferous tubule-like structure composed of SCLCs. Furthermore, conditioned medium prepared from TesLCs has a robust inducible activity to differentiate EpiLCs into PGCLCs. Our results demonstrate conditions for in vitro reconstitution of a testicular environment from ESCs and provide further insights into the generation of sperm entirely in xeno-free culture.

[1] Institute of Medical Sciences, Foresterhill, University of Aberdeen, Aberdeen, UK. [2] Institute of Ophthalmology, University College London, London, UK. ✉email: rsekido@abdn.ac.uk

Male infertility is a significant issue in animal reproduction. Even though assisted reproductive technology, e.g., in vitro fertilisation (IVF) and intracytoplasmic sperm injection (ICSI), overcomes some male infertility, the majority of germ cell deficiencies remain untreatable. Recent advances in stem cell technologies shed light on the availability of pluripotent stem cells as another source of germ cells.

It is already possible to derive primordial germ cell-like cells (PGCLCs) and spermatogonial stem cell (SSC)-like cells via epiblast-like cells (EpiLCs) from embryonic stem cells (ESCs) or induced pluripotent stem cells (iPSCs)[1–3]. However, these cells are not able to differentiate into sperm in vitro, suggesting lack of proper testicular environment in present culture conditions. Sertoli cells or Sertoli cell-derived signalling molecules are the candidate for the environmental factors because they physically interact with germ cells in vivo. Sertoli cells originate from the coelomic epithelium of the mesonephros[4], which is derived from mesoendoderm (ME) via intermediate mesoderm (IM). Sertoli cell precursors are specified by the expression of the Y-linked testis-determining gene, *Sry*, within a NR5A1/SF1-positive coelomic epithelial cell population[5]. Both SRY and SF1 directly bind to testis-specific enhancers upstream of the *Sox9* gene and synergistically activate its expression[6,7]. *Sox9* expression is maintained in Sertoli cells by a combination of its transcriptional autoregulation and the FGF9 signalling[6,8]. We previously created a transgenic mouse line having the enhanced cyan fluorescence protein (eCFP) gene driven by a *Sox9* testis-specific enhancer core elements (known as *tesco*), which elicits CFP expression exclusively in Sertoli cells[6]. We then established ESC lines from the transgenic mice, named *tesco-cfp* ECSs (hereinafter *tc*ESCs).

Previously, attempts had been made to generate artificial Sertoli cells from either fibroblasts or ESCs[9–13]. However, their physiological activities as functional Sertoli cells remained unanswered. In this study, we conducted sequential induction from *tc*ESCs to TesLCs including SCLCs and steroidogenenic cells. Subsequently, we created testicular organoids in aggregation co-culture of TesLCs and PGCLCs or EpiLCs.

## Results

**Generation of testicular somatic cell-like cells from mouse embryonic stem cells.** We performed a 3-step induction that consists of ESC-to-ME, ME-to-IM, and IM-to-SCLC[14]. *tc*ESCs were cultured in MEIM medium supplemented with combinations of a Wnt activator CHIR99021, a retinoic acid (RA) receptor agonist TTNPB, a RhoA/ROCK inhibitor Y27632 for the first two steps, and subsequently in RAG medium supplemented with CHIR99021, Y27632 and BMP7 for the third step (Supplementary Fig. 1a). Following the formation of aggregates for the first 2 days, ESCs started to differentiate into mesenchymal-shaped cells between day 5 and 8, and finally formed an epithelial layer with the basement membrane on its edge by day 12 (Supplementary Fig. 1b). Immunocytochemistry confirmed that SOX2-positive ESCs disappeared and PAX2-positive IM-like cells appeared within the first 5 days (Supplementary Fig. 1c). Sertoli cell markers SOX9 and GATA4 were observed by day 5 and day 12, respectively (Supplementary Fig. 1d). Quantitative RT-PCR analyses showed that the expression of ESC markers, *Sox2* and *Nanog*, declined immediately and then IM markers, *Lhx1*, *Osr1* and *Pax2*, as well as renal markers *Six2*, *Hoxd11* and *Foxd1*, were reciprocally activated by day 5 (Supplementary Fig. 1e). Subsequently, as the expression of IM and renal markers decreased after day 5, an adrenogonadal marker, *Cited2*, and steroidogenic markers, *Hsd3b1* and *Hsd17b3*, were progressively expressed by day 12 (Supplementary Fig. 1f). The occurrence of Sertoli cell differentiation was verified by the expression of five key transcription factors such as *Sox9*, *Wt1*, *Gata4*, *Sf1* and *Dmrt1*, previously defined by Buganim et al.[9] in their studies of fibroblast-to-SCLC induction. The expression levels of *Sox9*, *Wt1* and *Gata4* were robustly increased and those of *Sf1* and *Dmrt1* moderately (Supplementary Fig. 1g). The expression of *Sf1* and *Sox9* peaked by day 2 and day 5, respectively, which may mimic the lag observed during Sertoli cell differentiation in vivo[6]. These results suggested that both SCLCs and potential steroidogenic cells were present in this cell population, thereby referred to testicular somatic-like cells (TesLCs).

However, a major concern was a higher rate of cell death than cell proliferation in the ME-to-IM step (Supplementary Fig. 1h). To overcome it, we skipped this step by placing the cells directly into TesLCs induction under four conditions; I, II, III and IV, using different combinations of FGF9, BMP7 CHIR99021 and TTNPB (Supplementary Fig. 2a). Y27632 was then removed from the media because no obvious effect on the cell survival rate was observed. CFP expression was efficiently induced in the presence of CHIR99021, i.e. condition III and IV (Supplementary Fig. 2b). Interestingly, there were less CFP-positive cells in the presence of TTNPB, suggesting that RA signalling antagonizes gonadal cell differentiation[15]. The most efficient appearance of CFP-positive cells was achieved in condition III with CHIR22091, BMP7 and FGF9 altogether (hereinafter CBF treatment) (Fig. 1a), although CFP expression levels seemed to be downregulated after day 10 (Fig. 1b), presumably because not only did SCLCs become less proliferative, but continuous activation of Wnt signalling by CHIR99021 may have resulted in the transdifferentiation of SCLCs to ovarian granulosa cell-like cells[16–18]. In this 2-step induction, following the generation of IM-like cells by day 5 (Fig. 1c, top), *tc*ESCs were differentiated into TesLCs by day 8. The presence of steroidogenic cells and SCLCs was verified by the expression of HSD3B (Fig. 1c, bottom) and the colocalization of CFP and SOX9 or SF1 and SOX9, respectively (Fig. 1c, middle). Quantitative RT-PCR showed that the expression of *Sox2* and *Nanog* immediately decreased within the first 6 days whereas renal markers, *Foxd1* and *Hoxd11*, a steroidogenic gene *Hsd3b1* and the five Sertoli cell transcription factors, except for *Dmrt1*, were activated although the activation of *Foxd1*, *Hoxd11* and *Sf1* was transient (Fig. 1d). These results were consistent with those in 3-step induction.

For efficient induction, GDNF and IGF1 were added to CBF (i.e. CBF + IG treatment) because they are known to stimulate the proliferation of Sertoli cells[19–21]. The CBF + IG treatment not only allowed CFP expression to be maintained for as long as day 20, but also facilitated the formation of a ridge-like structure (Fig. 1e). The number of Ki67-positive proliferative cells evidently increased (Fig. 1f). In the ridge, SCLCs maintained the expression of SOX9 and GATA4, and laminin-positive basement membrane underlay the SCLCs (Fig. 1g).

**Facilitation of the formation of tubular-like structures by overexpression of Nr5a1/Sf1 in ESCs.** Despite CBF or CBF + IG treatment, *Sf1* expression remained low beyond day 2. It is known that *Sf1* is required for formation of the gonad and multiple steps in the generation of SCLCs and formation of tubular-like structures[4,22]. The evidence prompted the speculation that forced expression of *Sf1* in *ts*ESCs might robustly enhance SCLC differentiation. Of several created stable lines constitutively expressing *Sf1*, one (so called *tc;Sf1*ESCs) was found to express *Sf1* at levels ~100-fold greater than in the parental *tc*ESCs, (Fig. 2a). TesLCs derived from *tc;Sf1*ESCs, i.e., *tc;Sf1*TesLCs, became CFP-positive from day 7 to 8 until at least days 18–20 with CBF treatment. Intriguingly, interconnected tubule-like structures appeared with outlining CFP expression

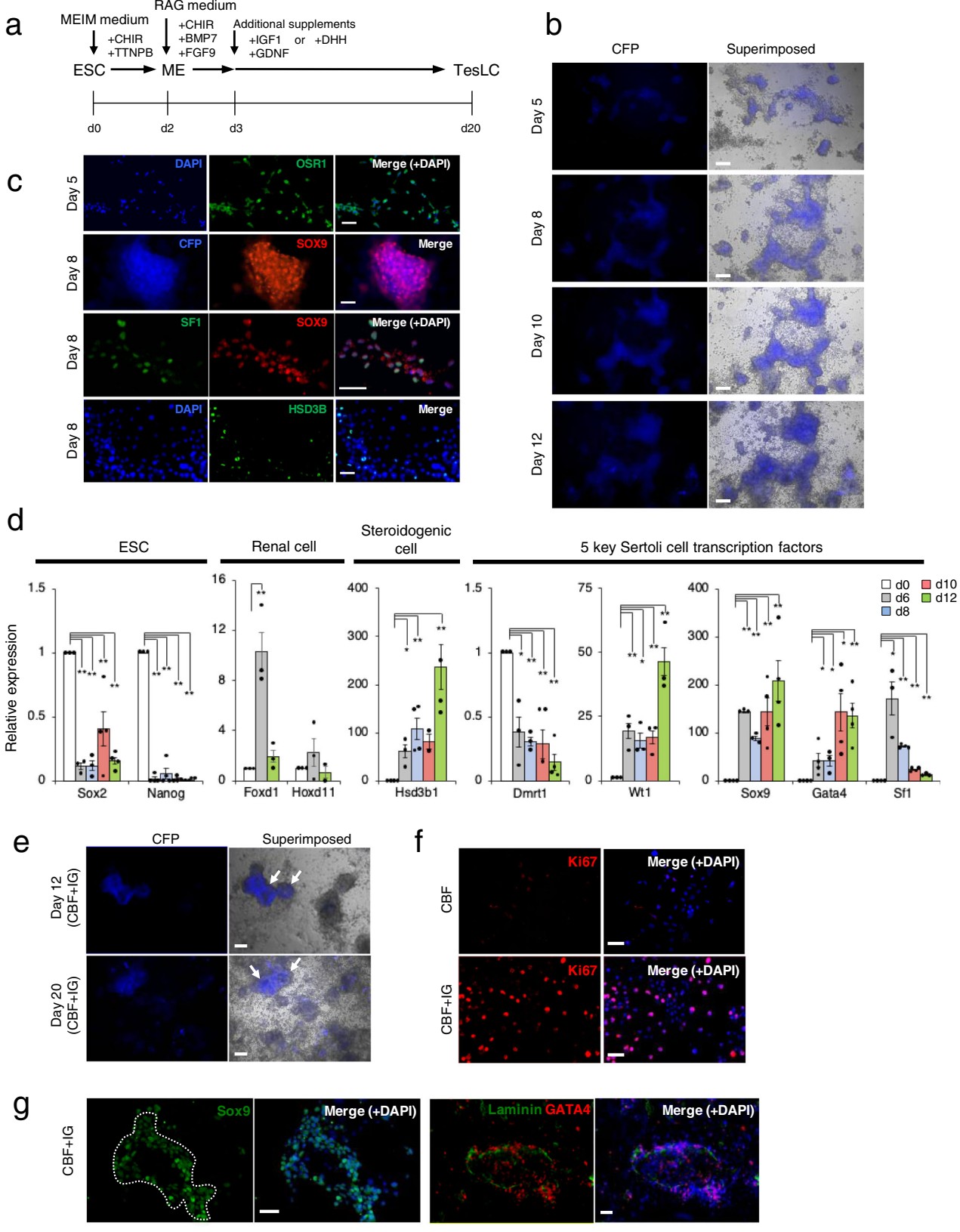

around day 18 (arrowheads in Fig. 2b,). Unlike in the case of *tc*ESCs, the administration of IGF1 and GDNF to *tc*;*Sf1*ESCs had no significant effect on the formation of the tubule-like structure. Hence, they were omitted from subsequent cultures. A subset of the cells constitutively expressing SF1 from the transgene was SOX9-positive (Fig. 2c, top) and almost all

SOX9-positive cells were positive for WT1 (Fig. 2c, bottom), confirming that the double-positive cells were SCLCs. The SCLCs were bounded by a laminin-positive basement membrane (Fig. 2d, top). A hollow tubule-like structure developed at later stage (Fig. 2d, bottom, arrows). The expression of the five Sertoli cell transcription factors, as well as markers for cells

**Fig. 1 Generation of TesLCs from ESCs. a** Schematic outline of TesLC differentiation with media containing CHIR99021, BMP7 and FGF9 (CBF), CBF + IGF1 + GDNF (CBF + IG) or CBF + DHH (CBF + D). **b** Bright-field images and those superimposed with CFP-fluorescent images of cells treated with CBF at different time-points during differentiation. **c** Molecular characterisation during TesLC differentiation in CBF-treated culture. The expression of an IM marker (OSR1) was detected at day 5. CFP expression and Sertoli cell markers (SF1/NR5A1 and SOX9) colocalized at day 8. A steroidogenic marker (HSD3B) also appeared at day 8. **d** Quantitative RT-PCR analysis for the expression of ESC markers (*Sox2* and *Nanog*), renal markers (*Foxd1* and *Hoxd11*), a steroidogenic marker *Hsd3b1*, and 5 key transcription factors required for Sertoli cell differentiation (*Sox9, Gata4, Wt1, Dmrt1* and *Sf1/Nr5a1*), over 12 days of differentiation. Fold expression changes relative to day 0 were calculated by ΔΔCt method with standard error. Three or four biological replicates, each of which had three technical replicates, were used for two-sided *t*-test. \*\**P* < 0.01, \**P* < 0.05. **e** Bright-field and CFP-superimposed f images of cells treated with CBF + IG at day 12 and 20. Arrows indicate ridge-like structures. **f** Immunohistochemistry detected an increase of Ki67-positive proliferative cells (red) treated with CBF + IG (bottom row), compared with only CBF treatment (top row) at day 8. **g** A vast majority of CBF + IG-treated cells were SOX9-positive (green). The dotted line demarcates a ridge-like structure. GATA4-positive cells (red) were surrounded by basal membrane denoted by laminin expression (green) at day 8. DAPI (blue) for nuclear staining. Scale bar, 50 μm.

committed to Sertoli cells, such as *Vnn1, Fshr1* and *Amh*, was elevated (Fig. 2e).

In *tc;Sf1*TesLCs culture, several genes encoding steroidogenic enzymes including *Star, Cyp17a1* and *Hsd3b6*, were also progressively activated up to day 20 even without cAMP treatment (Fig. 2f). Intriguingly, a number of the cells were double-positive for GATA4 and HSD3B or StAR (Fig. 2g), which may represent characteristics of multipotent gonadal precursor cells from which Sertoli cells and Leydig cells originate[23]. During gonadogenesis, Sertoli cells and Leydig cells in their immature or foetal form are replaced with those in mature or adult form, respectively. Of particular interest is HSD17B3, the crucial enzyme that converts androstenedione to testosterone in the canonical androgenic pathway, which is expressed in immature Setoli cells, instead of foetal Leydig cells[24]. Later, it is expressed in adult Leydig cells, but no longer in mature Sertoli cells. In contrast to 2-fold activation of *Hsd17b3* expression in the 3-step induction (Supplementary Fig. 1f), little if any *Hsd17b3* activation was observed in the 2-step induction. The results suggested inefficient differentiation of immature Sertoli cells or adult Leydig cells. Especially, the former coincided with a low expression level of desert hedgehog (*Dhh*) that is secreted from immature Sertoli cells (Fig. 2e). The fact that DHH is required for the specification of Leydig cells within the gonadal interstitial cell population in vivo[25], prompted us to test whether the administration of DHH to CBF (i.e. CBF + D) could facilitate Leydig cell differentiation. Indeed, significant activation of steroidogenic markers was observed, along with a dramatic reduction of the expression of a Sertoli cell marker, *Sox9*, and an undifferentiated interstitial cell marker, *Coup-tfII*, (Fig. 2h). Enzyme-linked immunosorbent assay (ELISA) assays for secreted testosterone levels in media revealed that *tc;Sf1*TesLCs produced ~400 pg/ml of testosterone when treated with CBF + D treatment for day 20 (Fig. 2i).

Taking advantage of CFP-fluorescence in SCLCs, we performed flow cytometry to measure the induction efficiency of SCLCs. The analysis revealed that viable SCLCs were generated with a frequency of <0.5% from *tc*ESCs and 3.7% from *tc;Sf1*ESCs (Supplementary Fig. 3a). However, because the resistance of TesLCs to enzymatic reaction under non-harmful condition lowered the dissociation efficiency, the overall induction efficiency may increase when dissociating condition is further optimised. The expression of key Sertoli cell transcription factors was enriched in the CFP-positive population although significant levels of *WT1* expression were also detected in the CFP-negative population (Supplementary Fig. 3b, c).

Collectively these results suggest that sequential treatments with CHIR99021 and TTNPB, followed by CHIR99021, BMP7 and FGF9, induce the differentiation of ESCs into TesLCs that consist of mixed cell-types including SCLCs, steroidogenic cells and interstitial cell-like cells. In addition, *Sf1*-overexpression enhances not only the induction of mature SCLCs within the

TesLC population, but the differentiation of testosterone-producing cells in the presence of DHH.

**Generation of testicular organoids in aggregation co-culture of testicular somatic-like cells and epiblast-like cells.** We suppose that physical interaction between PGCLCs and SCLCs is essential to proceed with in vitro spermatogenesis as it takes place in vivo. Indeed, it was shown that PGCLCs acquired a competence to develop into spermatogonial stem cells by aggregating with somatic cells isolated from E12.5 mouse testes in co-culture[2]. We therefore performed co-culture of PGCLCs and TesLCs (Supplementary Fig. 4a). In accordance with the standard protocol described by Hayashi et al.[1], PGCLCs were generated from *Prdm1-gfp* ESCs via EpiLC differentiation denoted by FGF5 expression (Supplementary Fig. 4b). Then PGCLCs from day 6 of culture (d6PGCLCs) and *tc;Sf1*TesLCs from day 12 (d12*tc;sf1*TesLCs) were dissociated. In all, 1500 cells from each dissociation were aggregated in RAG medium containing CHIR99021 (RAG + C) supplemented with SSC factors such as GDNF, bFGF, EGF and LIF. Although clusters of GFP-positive PGCLCs and CFP-positive SCLCs appeared adjacent to each other, seminiferous tubule-like arrangement did not occur (Supplementary Fig. 4c) by day 6 of co-culture. Almost all GFP-positive cells expressed mouse *vasa* homologue (*Mvh*), also known as *Ddx4*, suggesting that efficient induction of PGCLCs had been achieved (Supplementary Fig. 4d). The expression of *Sox9* and *Sf1*, as well as an early PGC marker *cKit* and a late PGC/gonocyte marker *Dazl*, was upregulated (Supplementary Fig. 4e, f), indicating that PGCLCs and SCLCs were maintained in the aggregates. However, genes characteristic to SSCs, *Plzf, Id4* and *Gfra1*, were not activated (Supplementary Fig. 4f). To investigate whether meiosis could be induced, aggregates were treated with a combination of RA and testosterone (TS)[26]. In addition to consistent *Mvh* expression, early meiotic markers such as *Stra8* and *Sycp3*, and a haploid marker *Haprin* were highly upregulated in the presence of RA, but not TS. No synergistic action was observed (Supplementary Fig. 4g).

Several lines of evidence suggest that Sertoli cells or conditioned media of testicular somatic cells facilitate the differentiation of human ESCs into PGCLCs[27–29], despite the fact that the interaction between ESCs and Sertoli cells do not occur in vivo. To examine whether TesLCs have a similar activity, we co-cultured EpiLCs and TesLCs in the RAG + C medium with no other supplement (Fig. 3a). In contrast that co-culture of EpiLCs with HEK293T cells failed to produce either distinct aggregates or GFP expression (Fig. 3b), EpiLCs and d8*tc*TesLCs formed a three-dimensional structure in which GFP-positive PGCLCs and CFP-positive SCLCs appeared by day 6 (henceforth referred to as testicular organoid) (Fig. 3c). The results indicate that the PGCLC differentiation requires physical contact with specific testicular cells. SOX9-positive

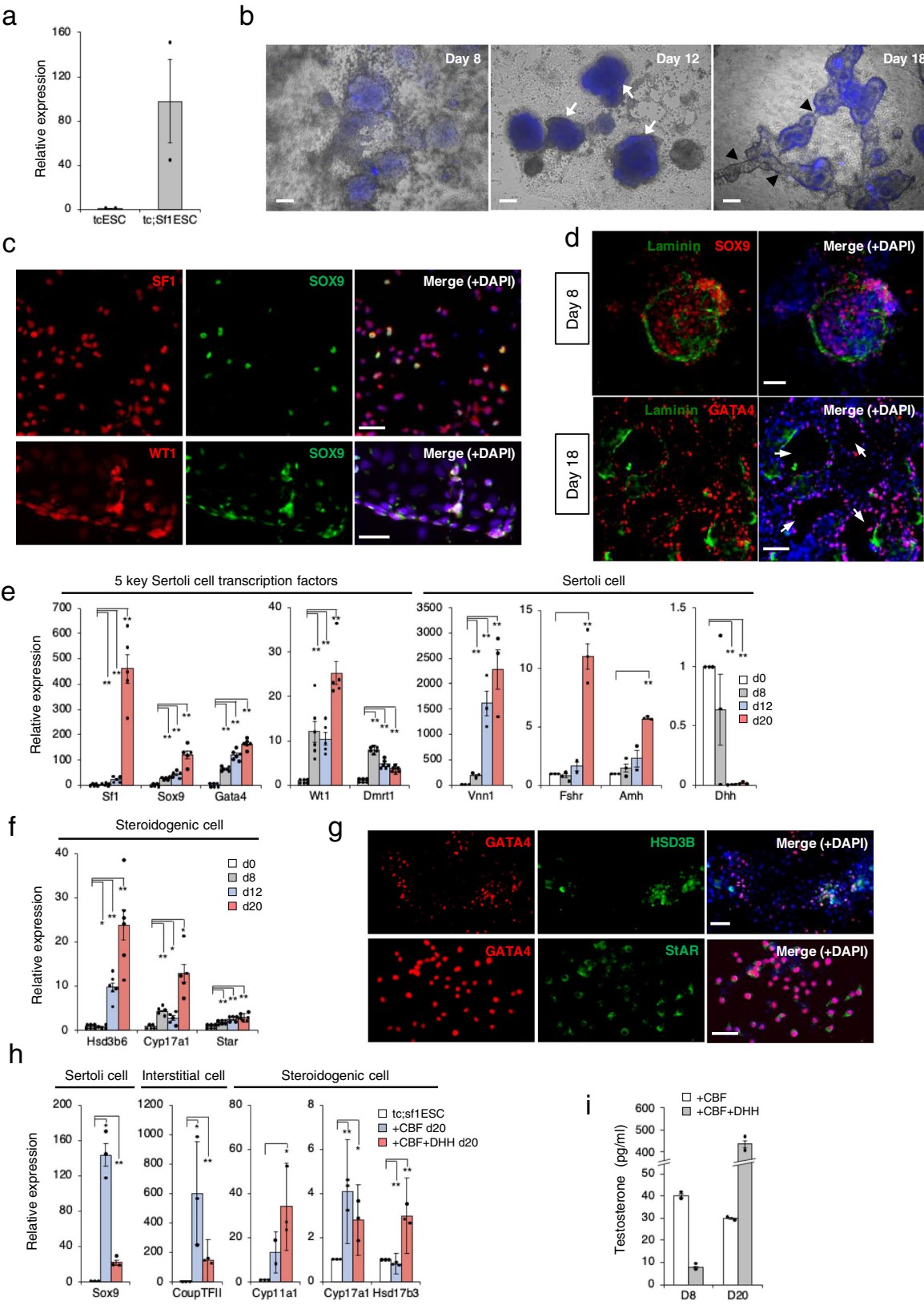

cells and MVH-positive cells localised at marginal region of the organoid, although no overt structure analogous to the seminiferous tubules was observed (Fig. 3c). Intriguingly, we found that co-culture of EpiLCs and d8*tc;Sf1*TesLCs self-organised into testicular organoids with evident seminiferous tubule-like structures from day 6 of culture, and especially GFP-positive PGCLCs were encapsulated by CFP-positive SCLCs (Fig. 3d). The testicular organoid-derived PGCLCs, or tod-PGCLCs, were MVH-positive and resided on the edges of the seminiferous tubule-like structure where SOX9-positive SCLCs were bounded by a layer of laminin (Fig. 3d). Flow cytometry analysis revealed that 2% of the organoid cells were GFP/CD49f-double positive PGCLCs (Supplementary

**Fig. 2 The effect of *Sf1* overexpression on TesLC differentiation. a** The *tc;Sf1*ESC line expresses *Sf1* from a transgene as 100 times as much as the parental *tc*ESC line. *n* = 2 biological replicates, each of which had three technical replicates. **b** Superimposed CFP-fluoresce on bright-field images of CBF-treated *tc; Sf1*ESCs during TesLC differentiation. Arrows indicate tubule-like structures appeared at day 12 and arrowheads highlight interconnection of the tubules. **c** Co-immunostaining of SF1 (red), WT1 (red) and SOX9 (green) for detection of Sertoli cell differentiation in CBF-treated *tc;Sf1*ESCs at day 8. **d** Detection of basal membrane by immunostaining of laminin (green) and Sertoli cell markers, GATA4 (red) and SOX9 (red) at day 8 and 18. Arrows indicate tubule-like structures. **e**, **f** Relative expression of 5 key Sertoli cell transcription factors and committed Sertoli cell markers (**e**), and steroidogenic markers (**f**) at day 0, 8, 12 and 20 time-points during TesLC differentiation. Fold expression changes relative to day 0 were calculated by ΔΔCt method with standard error. Three to six biological replicates, each of which had three technical replicates, were used for two-sided *t*-test. \*\**P* < 0.01, \**P* < 0.05. **g** Immunohistochemistry for detection of a steroidogenic enzyme HSD3B (green) and StAR (green), co-stained with GATA4 (red) at day 8. **h** Effects of DHH on the expression of a Sertoli cell marker (*Sox9*), an interstitial cell marker (*Coup-TFII*) and steroidogenic cell markers (*Cyp11a1, Cyp17a1* and *Hsd17b3*) at day 20 of culture. Fold expression changes relative to *tc;Sf1*ESCs cultured in 2iLIF medium were calculated by ΔΔCt method with standard error. Three biological replicates, each of which had three technical replicates, were used for two-sided *t*-test. \*\**P* < 0.01, \**P* < 0.05. **i** Measurement of secreted testosterone in culture media by ELISA in the presence or absence of DHH. *n* = 3 technical replicates. DAPI (blue) for nuclear staining. Scale bar, 50 μm.

Fig. 3d). In the organoids, the expression of *Sox9* and *Sf1* was robustly activated by day 6 (Fig. 3e). A high level of *Dazl* expression and moderate levels of SSC markers, *Gfra1* and *Id4*, but not *Plzf*, were detected albeit *Stra8* activation remained as low as in the co-culture of PGCLCs and SCLCs (Fig. 3f).

In light of these results, we investigated whether the differentiation of SSC-like cells and subsequent meiosis took place within the organoids. Considering the possibility that predominant formation of an SSC niche environment for self-renewal inhibited proceeding with in vitro spermatogenesis[30–32], we expected that dissociation and re-aggregation of the organoids could allow PGCLCs to be released from the niche and to enter meiosis spontaneously (Fig. 3a). In the reaggregated, or secondary, testicular organoids high expression levels of *Mvh, cKit* and *Dazl* were maintained and *Id4* expression was also discerned, whereas expression of *Stra8* and *Sycp3* remained low or negligible (Fig. 3g). The administration of RA, TS or together, to the secondary organoids showed that only RA was able to activate *Mvh, Dazl, Stra8* and *Sycp3* (Fig. 3g and Supplementary Fig. 5).

Taken together, co-culture of EpiLCs and *tc;Sf1*TesLC developed a testicular organoid with the appearance of a seminiferous tubule-like structure composed of SCLCs and PGCLCs. A subpopulation of PGCLCs could have differentiated into gonocyte/SSC-like cells. RA administration to the testicular organoid upregulated the expression of SSC markers and even early meiotic markers.

**Primordial germ cell induction by conditioned media derived from testicular somatic cell-like cells.** The results prompted us to test the effect of TesLC-conditioned media (CM) on PGCLC induction. The CM was prepared from RAG + CBF medium used to culture *tc*TesLCs or *tc;Sf1*TesLCs for different durations, i.e. 8, 10, 12, 16 and 20 days. Each CM was administered to aggregated EpiLC cultures. After 6 days, PGCLC induction was observed in all of cultures containing CM derived from *tc;Sf1*TesLCs, whereas it declined in the cultures containing CM derived from *tc*TesLCs particularly maintained for 10 days or longer (Fig. 4a). The CM of d12*tc;Sf1*TesLCs robustly activated the expression of early/late PGC and meiotic markers including *Tcfap2c, Nonos3, Mvh, Stra8* and *Sycp3*, (Fig. 4b), although endogenous *Prdm1* expression peaked earlier than the others (Fig. 4c).

Flow cytometry analysis to determine PGCLC induction efficiency revealed that 18% of CM-treated cells were GFP/SSEA1 double positive PGCLCs, which was higher than those created by the standard protocol[1] or the testicular organoid method (Fig. 4d). As Murakami et al.[33] previously demonstrated that BMP4 is a key molecule to induce GFP expression in EpiLCs derived from *Prdm1-gfp* ESCs, we detected the activation of Bmp4 expression in both CFP-positive and negative populations within *tc*TesLCs and *tc;Sf1*TesLCs, although CFP-negative cells showed lower

expression in *tc;Sf1*TesLCs (Fig. 4e). This may due to the induction of factors antagonistic to the BMP pathway by *tc*TesLC-CM. The administration of LDN-193189, an inhibitor of the TGFβ/BMP signalling cascade, repressed GFP expression in a dose-dependent manner (Fig. 4f).

**Transcriptome of Sertoli cell-like cells and primordial germ cell-like cells in the testicular organoid.** We characterised the molecular basis for the differentiation of SCLCs and PGCLCs by high-throughput RNA-sequencing analyses. CFP-positive cells sorted from d12 *tc;Sf1*TesLCs were used as representative SCLCs. Global transcription profiles were compared between SCLCs, *tc*ESCs, *tc;Sf1*ESCs, and embryonic day 12.5 Sertoli cells (E12.5SCs) described by McClelland et al.[34]. We found 1813 differentially expressed genes (DEGs) between *tc;sf1*ESC and *tc*ESC (Supplementary Data 1a), ~6500 DEGs between SCLCs and *tc*ESCs, and between SCLCs and *tc;sf1*ESC, (Supplementary Data 1b, c), and over 8000 DEGs between E12.5SCs and *tc*ESCs, between E12.5SCs and *tc;Sf1*ESCs, and between E12.5SCs and SCLCs (Supplementary Data 1d–f). Clustering analysis and principal component analysis (PCA) revealed that SCLCs were distinct from ESCs and seemed to be still in the process towards fully differentiated Sertoli cells. (Fig. 5a, c). In total, 9 out of 1813 DEGs fitted in the 'Transcriptional regulation of pluripotent stem cells' (R-HSA-452723) term in Reactome Pathway Database (Supplementary Table 1a). Of over 6500 DEGs, 275 genes were categorised into gene ontology (GO) either 'Sex determination' (GO:0007530), 'Sex differentiation' (GO:0007548), 'Sertoli cell development' (GO:0060009) and 'Sertoli cell differentiation' (GO:0060008) (Supplementary Fig. 6a). Genes characteristic of Sertoli cells, notably the five transcription factors, *Ptgds, Vnn1, Gfra1, Gdnf, Insr* and *Cited2*, were upregulated in SCLCs, although their expression levels were lower than those in E12.5 Sertoli cells, except for a few genes (Fig. 5d and Supplementary Table 1b). Intriguingly, a significant upregulation of *Dmrt1* and *Fgf9* was observed only when compared between SCLCs and *tc; Sf1*ESCs, suggesting that a high level of *Sf1* expression enhanced SCLC differentiation. Furthermore, to explore sets of genes subjected to the SCLC differentiation, we performed GO enrichment analysis (Supplementary Data 2a–f). Forced *Sf1* expression in ESCs enriched gene sets associated with 'tissue development', 'cell differentiation' and 'cell proliferation' in an overexpression manner, whereas 'cell adhesion', 'ion transport' and 'neuron' in an underexpression manner. 'Reproduction' and 'extracellular components' were enriched as both over- and underexpressed gene sets. The enriched gene sets between *tc*ESCs and SCLCs were similar to those between *tc;Sf1*ESCs and SCLCs. Briefly, in addition to 'neuron' and 'neurogenesis' overrepresented in both over- and underexpression, 'cell morphogenesis' and 'cell death' were specifically associated with overexpression in

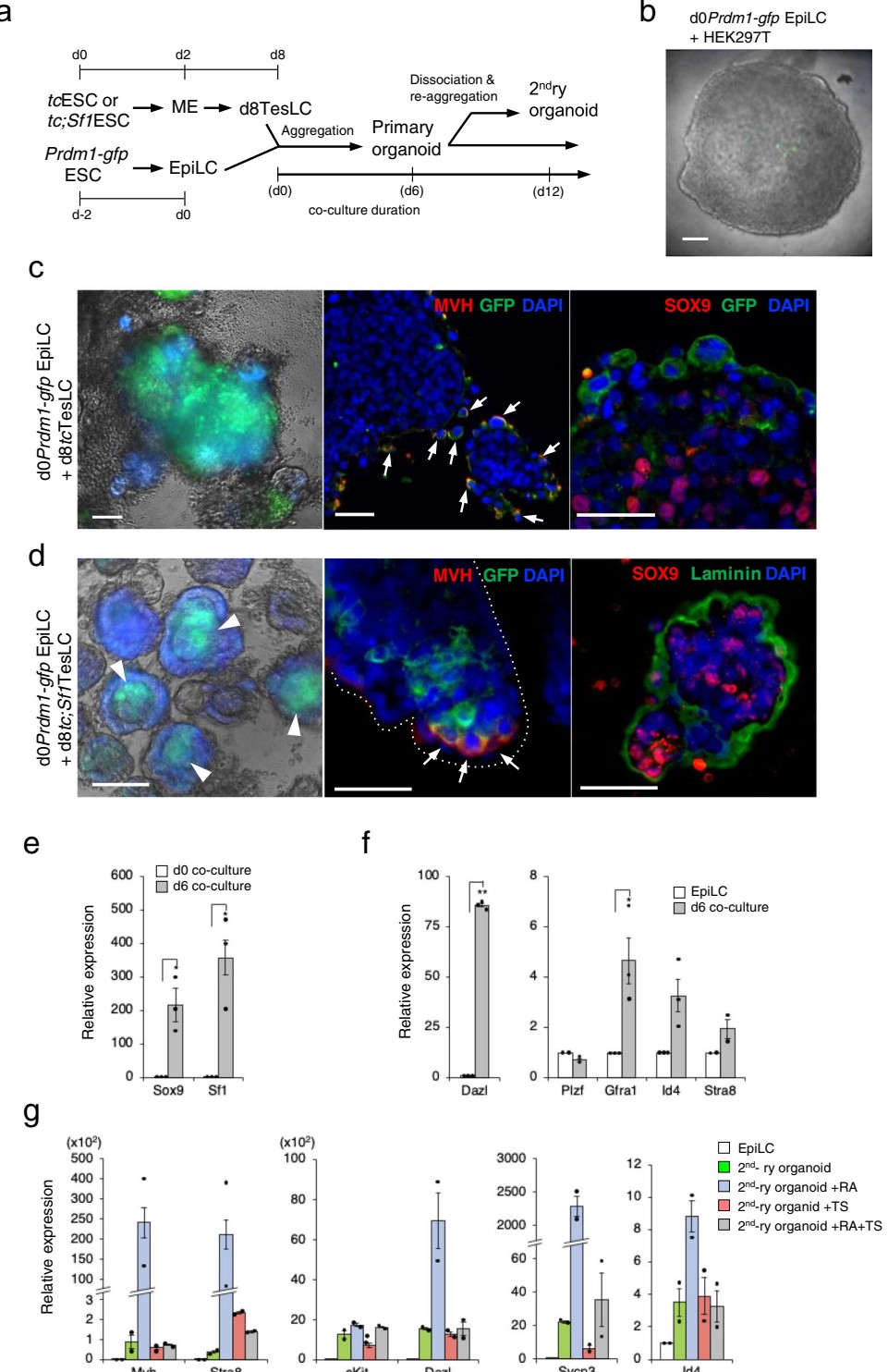

**Fig. 3 Formation of a testicular organoid. a** Schematic outline of testicular organoid formation by means of aggregation co-culture of d8TesLCs and EpiLCs. **b** Superimposed images of the aggregation of *Prdm1-gfp* EpiLCs with HEK293T cells. **c, d** Superimposed images of the aggregation of *Prdm1-gfp* EpiLCs with d8*tc*TesLCs (**c**) or d8 *tc;Sf1*TesLCs (**d**) at day 6 of culture and immunohistochemistry for SOX9 (red), MVH (red) laminin (green) and GFP (green). Arrowheads indicate PGCLCs embedded in a seminiferous tubule-like structure composed of SCLCs (**d**). Arrows indicate cells expressing both MVH and GFP. **e, f** Quantitative RT-PCR analysis for the expression of a Sertoli cell marker *Sox9*, a late PGC/gonocyte marker *Dazl*, SSC markers *Gfra1*, *Plzf* and *Id4*, and an early meiotic marker *Stra8* in day 6 of aggregate. Fold expression changes relative to day 0 (**e**) or parental EpiLCs (**f**) were calculated by ΔΔCt method with standard error. Two to three biological replicates, each of which had three technical replicates, were used for two-sided *t*-test. **\*\*P <** 0.01, **\*P < 0.05. g**, The effect of retinoic acid (RA) and testosterone (TS) on the expression of early PGC (*cKit*), late PGC (*Dazl* and *Mvh*), SSC (*Id4*) and early meiotic (*Stra8* and *Sycp3*) markers in the formation of the secondary aggregate. *n* = 2 biological replicates, each of which had three technical replicates. DAPI (blue) for nuclear staining. Scale bar, 50 μm.

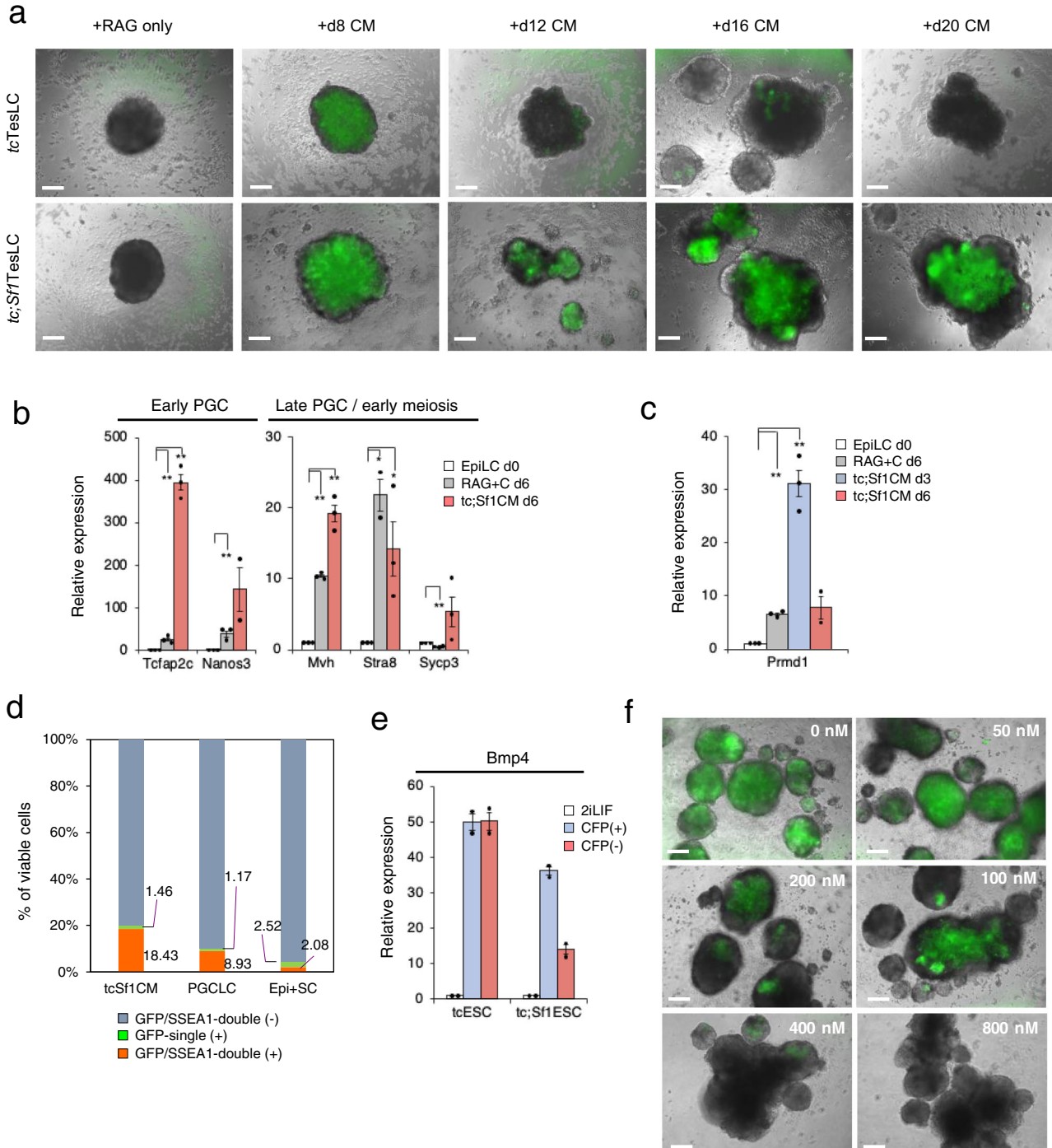

**Fig. 4 The effect of TesLC-conditioned medium on induction of EpiLCs to PGCLCs. a** PGCLC appearance after administration of conditioned media (CM) derived from CBF-treated *tc*TesLCs (top panels) and *tc;Sf1*TesLCs (bottom panels) with different durations of CBF-treatment. As a negative control, RAG medium only was used. Scale bar, 200 μm. **b, c** Relative expression of germ cell markers as indicated in EpiLCs treated with d8*tc;Sf1*TesLCs (*tc;Sf1*CM) for 6 days, compared to parental EpiLCs at day 0 of culture, was calculated by ΔΔCt method with standard error. Three biological replicates, each of which had three technical replicates, were used for two-sided *t*-test. **P < 0.01, *P < 0.05. **d** Higher efficiency of PGCLC induction by *tc;Sf1*CM, compared with that by the standard protocol (PGCLC) or EpiLC+SCLC co-culture (Epi + SC). Flow cytometry analysis with GFP and SSEA1-PE antibody was performed at day 6 of culture. The percentages of viable GFP/SSEA1 double- positive, GFP-single positive, and both negative cells are displayed. n = 4 biological replicates. **e** Relative expression of *Bmp4* in sorted CFP-positive SCLCs and CFP-negative cells, compared to ESC maintained in 2iLIF medium. n = 2 technical replicates. **f** Reduction of the number of PGCLCs by administration of LDN-193189 in a dose-dependent manner. Scale bar, 200 μm.

SCLCs, whereas DNA/RNA metabolism and mitochondrion with underexpression. These enriched gene sets were also consistent in the comparisons between *tc*ESCs and E12.5SCs and *tc;Sf1*ESCs and E12.5SCs. Finally, the comparison between SCLCs and E12.5SCs revealed that 'cytoplasmic components', 'intracellular transport', and 'apoptosis' were overexpressed in SCLCs, whereas 'ion channel', 'axon' and 'dendrite' as well as 'neuron' and 'neurogenesis' were underexpressed.

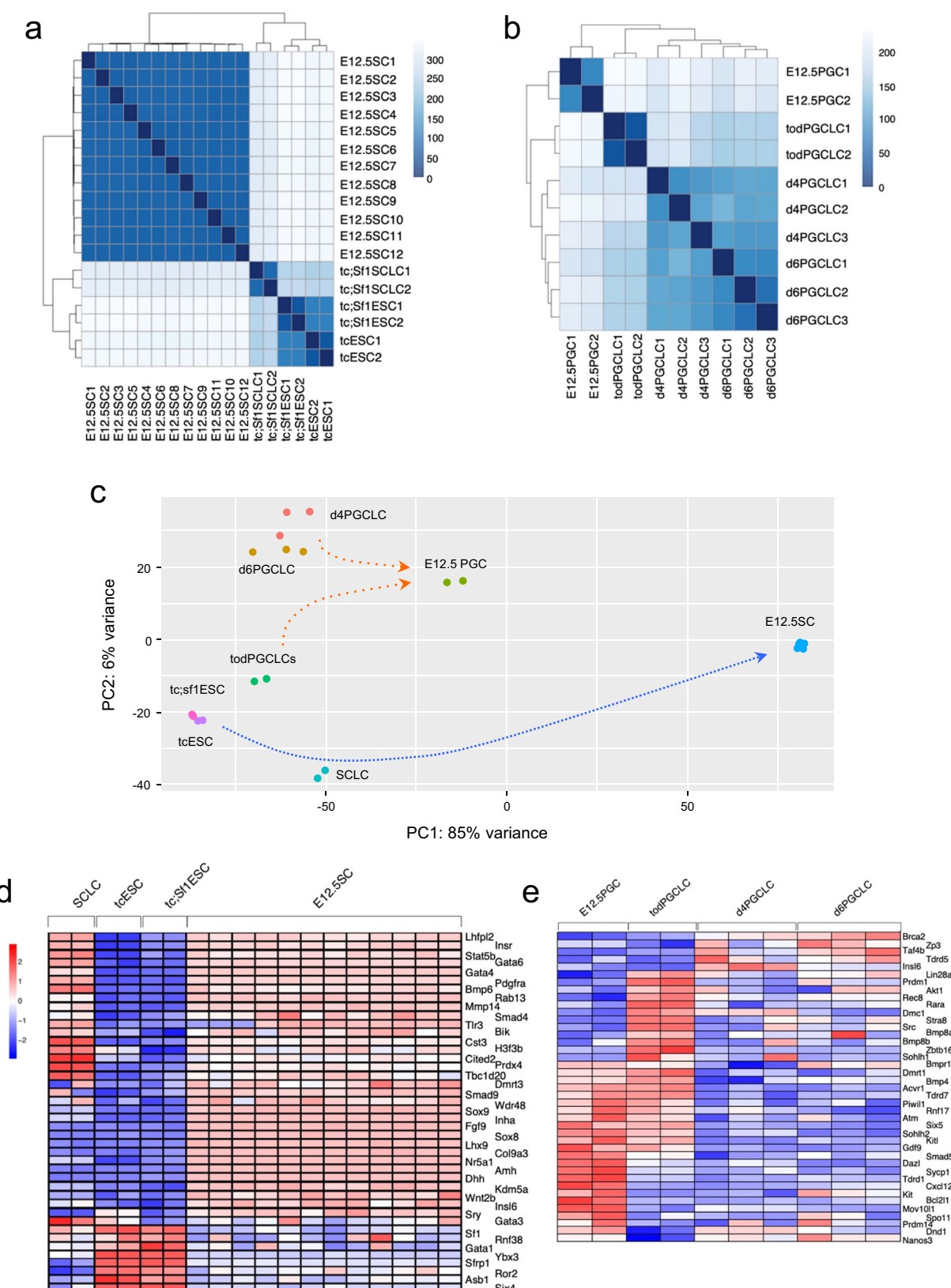

**Fig. 5 Global expression profiling during differentiation of ESCs into SCLCs and tod-PGCLCs. a, b** Sample distances among *tc*ESCs, *tc;Sf1*ESCs, *tc; Sf1*SCLCs and E12.5 Setroli cells (E12.5SC) (**a**), and among tod-PGCLCs, d4PGCLCs, d6PGCLCs and E12.5 PGCs (**b**). **c** PCA demonstrating changes in global gene expression during SCLC differentiation towards E12.5 Sertoli cells (blue dotted line) and the comparison between tod-PGCLCs, d4/d6PGCLCs and E12.5PGCs (green dotted line). **d, e** Heatmaps showing relative expression levels of a subset of GO-selected genes in SCLC differentiation (**d**) and PGCLC differentiation (**e**). The full sets of GO-selected genes are shown in Supplementary Fig. 6a, b.

For PGCLC differentiation, GFP/CD49f double-positive tod-PGCLCs sorted from day 6 testicular organoid cultures were compared with d4PGCLCs and d6PGCLCs previously generated by Hayashi et al.[1], as well as PGCs isolated from E12.5 mouse testis[35]. Clustering analysis and PCA showed a similarity of tod-PGCLC to d4/6PGCLCs (Fig. 5b, c). In all, 674 genes were categorised into either GO 'germ cell development' (GO:0007281) or 'male gamete generation' (GO:0048232) (Fig. 5e and Supplementary Fig. 6b). Selected representative genes during differentiation exhibited more or less similar levels of expression between tod-PGCLCs and d4/d6PGCLCs, except that *Klf4*, *Dmnt3b* and *Stra8* were upregulated more than 2-fold in tod-PGCLCs. The comparison between tod-PGCLCs and E12.5 PGCs revealed that ESC, ICM and epiblast markers remained higher in tod-PGCLCs than in E12.5 PGCs (Supplementary Table 1c).

## Discussion

We have demonstrated the generation of TesLCs from ESCs. TesLCs are heterogeneous cell population that contains SCLCs, steroidogenic cells and renal cells. However, the possibility that unknown cell types remain in the cell population cannot be ruled out. Additional treatment with IGF1 and GDNF facilitated the formation of a ridge-like structure, and a similar or more distinct tubule-like structure appeared when *Sf1* was overexpressed. As it is shown that SF1 and IGF signalling crosstalk during adrenal and testicular development[21,36], both factors may cooperate in terms of tubule formation. With respect to the derivation of Leydig cells from ESCs, it was demonstrated that overexpression of *Sf1* directs ESCs towards the steroidogenic lineage in the presence of sufficient amount of cAMP[37]. Hence, a high level of testosterone detected in *tc;Sf1*TesLC culture might have been released from if any Leydig cell-like-cells even though no cAMP was administered. Our results suggest that DHH treatment inclines to the differentiation of interstitial cell-like cells into testosterone-producing cells rather than SCLCs. As recent studies in *Hsd17b3* knockout mice have revealed that there is an *Hsd17b3*-independent testosterone-production pathway[38], testosterone may not be necessarily produced by SCLCs or Leydig cell-like cells, but any other unknown steroidogenic cells, notably multipotent gonadal precursor-like cells recognised as GATA4 and HSD3B/StAR double-positive cells. However, further characterisation is required to verify the existence of Leydig cell-like cells or such unknown steroidogenic cells.

We performed high-throughput transcriptome profiling to describe molecular characteristics of SCLCs and tod-PGCLCs. PCA and clustering analyses reveal similarities and/or divergence between samples. The wide divergence between SCLCs and E12.5SCs may be attributed to heterogeneity of SCLCs even after cell sorting because we used only CFP fluorescence for flow cytometry. Use of double or more markers may allow us to sort purer SCLCs. GO enrichment analysis revealed overrepresented gene sets in both upregulated and downregulated DEGs during SCLC differentiation. Overrepresented GOs in the differentiation of *tc*ESCs into SCLCs or *tc;Sf1*ESCs into SCLCs mostly overlapped those into E12.5SCs, reconfirming that SCLCs and Sertoli cells are in a similar differentiation process. Surprisingly, GOs associated with 'neuron' and 'neurogenesis' were highly enriched in comparisons even between ESCs and E12.5SCs, and between SCLCs and E12.5SCs. This may reflect similarities in molecular signatures between gonadogenesis and neurogenesis, both of which can be subjected to sex chromosome complement and sex hormones.

The generation of TesLCs allows us to create a testicular organoid. To our surprise, seminiferous tubule-like arrangement was efficiently formed by co-culturing TesLCs and EpiLCs, rather

than PGCLCs. In the latter case, a proper proportion of TesLCs and PGCLCs might be required and/or media should be optimised. The tod-PGCLCs did not undergo the completion of meiosis in the current condition. Zhou et al.[26] claimed that sequential treatment with morphogens and sex hormones in co-culture of PGCLCs and neonatal gonad-derived somatic cells recapitulated hallmarks of meiosis in vitro. However, a similar treatment was not sufficient to proceed with meiosis in our culture system. We also revealed that TesLC-conditioned medium has strong PGCLC-inducing activity when administered to EpiLCs. In contrast to the fact that somatic cells freshly prepared from pre/post-natal mouse gonads are required for most PGCLC cultures[3,25], the use of TesLCs or the conditioned medium is suitable not only for xeno-free culture, but also to avoid sacrificing foetus/pups to obtain gonadal somatic cells. Therefore, the present studies can be applied to other animal species including humans, especially for development of new assisted reproductive technology and therapeutic approaches to male infertility.

## Methods

**TesLC culture.** Mouse ESCs were maintained under feeder-free and serum-free conditions in N2B27 2iLIF medium containing 1000 U/ml of ESGRO mLIF (Millipore) and two inhibitors, i.e., 1 μM PD0325901 (Sigma Aldrich) and 3 μM CHIR99021 (Cayman Chemical), as previously described[39]. TesLC differentiation were performed according to the protocol described by Araoka et al. with modifications. Two media were prepared for the differentiation. The mesoendoderm/intermediate mesoderm (MEIM) medium contains DMEM/F12 + Glutamax (Gibco), 1x B27 supplement minus vitamin A (Gibco) and 1x penicillin-streptomycin (Gibco). The renal/adrenal/gonadal (RAG) medium contains DMEM/F12 + Glutamax, 1x B27 supplement minus vitamin A, 1x penicillin-streptomycin, 1x non-essential amino acids and 0.1 mM β-mercaptoethanol (Sigma Aldrich). For three steps of induction, ESCs were plated onto 25 ng/ml Synthemax-II sc substrate (Corning) and cultured in the MEIM medium supplemented with 10 μM Y27632 (Tocris), 1 μM TTNPB (Sigma Aldrich) and 3 μM CHIR99021 (Sigma Aldrich) for 2 days. The resulting ME-like cells were differentiated into IM-like cells in the MEIM medium supplemented with only 3 μM CHIR99021 for 3 days. The IM-like cells were then differentiated into gonadal cells in RAG medium supplemented with 10 μM Y27632, 3 μM CHIR99021 and 100 ng/ml BMP7 (PeproTech). For two steps of induction, the process of ME induction is the same as the one in the three steps. After 2 days of ME induction, the medium was replaced with RAG medium supplemented with different combinations of 1 μM TTNPB, 3 μM CHIR99021, 100 ng/ml BMP7 and 100 ng/ml FGF9 (PeproTech), and additionally 20 ng/ml GDNF (PeproTech), 100 ng/ml IGF1 (PeproTech) and 100 ng/ml DHH (BioLegend) as indicated. All the media components were described in Supplementary Table 2.

To obtain conditioned media, RAG medium was removed from TesLCs were cultured in RAG medium with CHIR99021, BMP7 and FGF9 for different durations as indicated. The media were collected and centrifuged to remove debris, and stored at −20 °C until use. Prior to using the media, they were diluted 1:2 in unconditioned fresh RAG medium. For BMP inhibition experiments, 50–800 nM of a type 1 BMP receptor inhibitor LDN-1931189 (Cayman Chemical) was added to 1:2 conditioned media preparations.

**Differentiation of EpiLCs and PGCLCs.** Differentiation was accomplished following the protocols described by Hayashi et al.[1]. To gauge the success of germ cell differentiation, the *Prdm1*-eGFP reporter line was used[40]. For EpiLC differentiation, *Prdm1*-eGFP ESCs were plated at a density of $2.5 \times 10^4$/cm² onto 12-well plates pre-coated with fibronectin (16.7 μg/ml). Differentiation medium consisted of N2B27 medium containing 12 ng/ml bFGF (PeproTech), 20 ng/ml activin A (PeproTech) and 1% KSR (Gibco). Cells were incubated at 37 °C and 5% CO₂ for 1.5–2 days. For PGCLC induction, $2-4 \times 10^4$ EpiLCs were plated onto round-bottom 96-well plates pre-coated with Lipidure (Amsbio) to prevent attachment. PGCLC differentiation medium consisted of GMEM (Gibco) supplemented with 15% KSR, penicillin–streptomycin, 2 mM L-glutamine, 0.1 mM non-essential amino acids, 1 mM sodium pyruvate, 0.1 mM β-mercaptoethanol (Sigma Aldrich), 500 ng/ml BMP4 (PeproTech), 1000 U/ml LIF (Millipore), 100 ng/ml SCF (PeproTech) and 50 ng/ml mouse EGF (PeproTech). Aggregated cells were monitored for fluorescence over the course of 6 days at 37 °C and 5% CO₂.

**Aggregation culture and organoid formation.** 1,500 each of d6 PGCLCs and d12 *tc;Sf1*TesLCs were plated onto round-bottom 96-well plates with Lipidure-coating and maintained in RAG medium containing 3 μM CHIR99021 (RAG + C) supplemented with 12 ng/ml bFGF (PeproTech), 20 ng/ml GDNF, 50 ng/ml mouse EGF and 1000 U/ml LIF. In all, 1 μM retinoic acid (Sigma Aldrich) and/or 10 μM testosterone (gift from Paul Fowler) were also supplemented when

needed. For aggregation of SCLCs and EpiLCs, 1500 each of these cells on round-bottom 96-well plates without Lipidure-coating were cultured in RAG + C medium with no supplement or 1 μM retinoic acid and/or 10 μM testosterone. For the secondary organoid formation, the primary organoids were dissociated with a mixture of Accutase (Stem Cell Technologies), 0.1% collagenase IV (Sigma Aldrich) and 0.1% dispase (Sigma Aldrich), and reaggregated in the same condition. Cultures were monitored for aggregate formation and fluorescence for up to 12 days. Imaging was captured by either Nikon Eclipse TE 2000E or Zeiss Axio Observer Z1 inverted fluorescence microscope and then processed by using ImageJ or Zen software.

**Transfection**. For the creation of *tesco-cfp;Sf1* ESCs, mouse *tesco-cfp* ESCs were electroporated with 2 μg of pCDNA3-*Sf1*[6] using mouse ES cell nucleofector solution and programme A-30 installed in Nucleofector II instrument (Lonza). Cells were immediately plated into fresh N2B27 2iLIF medium and left to incubate at 37 °C and 5% $CO_2$. After 24 h of recovery, 180 μg/ml of G418 was added to the culture medium. After 5 days of incubation, surviving non-fluorescent single colonies were picked up and expended. To generate ESCs constitutively expressing CFP, wild-type ESCs derived from C57BL6 mouse strain were electroporated with 2 μg of pCAGGS-eCFP under the same condition as *tesco-cfp* ESCs. The resulting CFP fluorescent ESCs were used as a positive control for SCLC sorting by flow cytometry.

**Quantitative RT-PCR**. Cells were lysed and RNA extracted using TRIzol reagent (Invitrogen) or RNeasy mini kits (Qiagen) according to manufacturer instructions. For impure samples, an additional round of purification using sodium acetate and ethanol precipitation was performed. According to manufacture's protocol, random hexamer-primed cDNA libraries were generated from 1 μg of total RNA with SuperScript II reverse transcriptase (Invitrogen). Quantitative RT-PCR reactions were performed using StepOnePlus real-time PCR instrument (Applied Biosystems). A 10 μl reaction was set up for each sample, which consisted of 1 μl of diluted cDNA, 5 μl of 2x SYR Green reaction mix (Applied Biosystems), 0.5 μl of 2 μM forward primer, 0.5 μl of 2 μM reverse primer and 3 μl of nuclease-free water (Ambion). Samples were run in duplicate or triplicate. Housekeeping gene *Gapdh* was used to normalise results. All primers used were described in Supplementary Table 3.

**Immunohistochemistry**. In preparation for immunocytochemistry, cells were fixed in 4% PFA for 15 min at room temperature and then permeabilised for 1 hr with 0.1% Triton X-100 in PBS. After blocking with 3% BSA, samples were incubated with primary antibodies for 1 h at room temperature or overnight at 4 °C, followed by reaction with fluorescent secondary antibodies for another hour. Antibodies used were rabbit polyclonal anti-active caspase 3 (1:500, Abcam, ab2302), rabbit anti-FGF5 (1:50, Proteintech, 18171-1-AP), goat polyclonal anti-GATA4 (1:200, SantaCruz, sc-1237), chicken polyclonal anti-GFP (1:500, Abcam, ab13970), rabbit polyclonal anti-HSD3B (1:500, gift from Ian. Mason), rat polyclonal anti-Ki67 (1:500, BioLegend, 652401), rabbit polyclonal anti-laminin (1:1000, gift from Harold Erickson), rabbit polyclonal anti-MVH (1:300, Novus Biologicals, NBP2-24558), rabbit polyclonal anti-OSR1 (1:200, Abcam, ab230627), anti-Pax2 (1:200, Novus Biologicals NPB2-57700), goat polyclonal anti-SOX2 (1:200, Novus Biologicals, AF2018), goat anti-SOX9 (1:200, Novus Biologicals, AF3075), rabbit polyclonal anti-SF1/NR5A1 (1:200, Novus Biologicals NPB1-52823), rabbit polyclonal anti-STAR (1:200, Biorbyt, orb7014), rabbit polyclonal anti-STRA8 (1:200, Abcam, ab49602) and rabbit anti-SYCP3 (1:200, Novus Biologicals, NB300-230).

**Enzyme-linked immunosorbent assay**. Testosterone was extracted from culture media using a protocol provided by Testosterone ELISA kit (Enzo Life Sciences). Briefly, 400 μl of culture media were mixed with 400 μl of diethyl ether and left for 5 min to allow layers to separate. The upper ether supernatant was transferred to a fresh tube. The extraction was repeated twice and the ether layers combined. After desiccated, testosterone was stored at −20 °C. The testosterone samples were reconstituted in the Assay Buffer provided and the optical density at 405 nm was measured by PHERAstar FSX microplate reader. All the media samples and control testosterone standards were run in triplicate.

**Flow cytometry**. For controls, FITC- and Pacific Blue-coated beads were used to compensate fluorescence between GFP and CFP, respectively. For each experiment, unstained wild-type (WT) cells were differentiated alongside reporter lines. Prior to flow cytometry, cells were washed with PBS and dissociated with a mixture of Accutase (Stem Cell Technologies), 0.1% collagenase IV (Sigma Aldrich) and 0.1% dispase (Sigma Aldrich). Cells were re-suspended in 1x PBS, 2 mM of EDTA and 2% of BSA. After reacting with rat monoclonal CD49f-PE (1:11, Miltenyi Biotech) or human monoclonal SSEA1-PE (1:50, Miltenyi Biotech) antibodies and a viability dye (DAPI or propidium iodide), flow cytometry was performed using BD LSR II flow for sorting, cells were collected in FBS-coated tubes containing 100–500 μl of buffer. RNA was then extracted from each cell fraction using a Qiagen RNA Extraction Micro kit.

**High-throughput RNA sequencing**. In all, 12-day-old *tesco-cfp;Sf1* SCLCs and 6-day-old testicular organoids were prepared. RNA was extracted from the sorted CFP-positive SCLCs and GFP/SSEA1 double positive PGCLCs, as well as parental *tesco-cfp* and *tesco-cfp;Sf1* ESCs, by using The ARCTURUS® PicoPure® RNA Isolation Kit. The preparation of cDNA library and high-throughput sequencing were carried out by Eurofins Genomics in Germany. Briefly, RNA containing poly-A was selected and random primed strand-specific cDNA was synthesised. Following adapter ligation and adapter specific PCR amplification, a 50 bp pair-end sequencing on Illumina HiSeq4000 platform was performed. The sequence reads were filtered for low quality bases by using Trimgalore, and then were aligned to the mouse genome (version GRCm38/mm10) using StAR.

**Quantification of differential transcriptome**. Transcripts with statistical significant of Padj≤0.05 (adjusted *p*-value according to method adjust for multiple testing)[41] and the number reads bearing more than 10 were modelled for each sample. TPM values for each of the samples were calculated in order to compare expression levels across samples. Differential expression analysis was performed using DESeq2. Differentially expressed genes were identified as those having a fold change of greater than two and an adjusted *p*-value of <0.05. To compare our transcriptome data with those published in public domain, such as E12.5 embryonic Sertoli cells (SRR1783798 - 1783809), mPGCLCd4 (SRR4241909 - 4241911), mPGCLCd6 1,2,3 (SRR4241912 - 4241914) and E12.5 embryonic PGCs (SRR3105904 and 3105905), we obtained SRA files and processed to FASTQ reads. These reads were processed through the bioinformatics pipeline in a similar manner to our data.

**Gene ontology search and GO enrichment analysis**. EMBL-EBI QuickGO (https://www.ebi.ac.uk/QuickGO) and GSEA (https://www.gsea-msigdb.org/gsea) was used for GO search. Six GO terms; 'sex determination' (GO: 0007530), 'sex differentiation' (GO: 0007548), 'Sertoli cell development' (GO: 0060009) and 'Sertoli cell differentiation' (GO: 0060008) were selected for SCLC generation. Two GO terms; 'germ cell development' (GO:0007281) and 'male gamete generation' (GO: 0048232) were selected for tod-PGCLC generation. WEB-based GEne SeT AnaLysis Toolkit (WebGestalt) (http://www.webgestalt.org) was used for GO enrichment analysis. The analysis was performed for upregulated and downregulated genes that were defined by log fold change (LFC) > 1 or < −1 and adjusted *p*-value (Padj) ≤ 0.05.

**Reporting summary**. Further information on research design is available in the Nature Research Reporting Summary linked to this article.

## Data availability

RNA-seq data that support the findings of this study have been deposited in the NCBI Gene Expression Omnibus (GEO) with the GEO accession codes "GSE149932". All other data are available from the corresponding author on reasonable request. Data of graphs in the main figures are provided as Supplementary Data 3. All relevant data are available from the corresponding author on reasonable request.

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

## Acknowledgements

We thank Dr Robin Lovell-Badge and Prof Azim Surani for *tesco-cfp* ESC line and *Prdm1-gfp* ESC line, respectively. We also thank the University of Aberdeen Microscopy, Histology and real-time PCR facilities as well as the Iain Fraser Cytometry Centre for training and use of their equipment. H.R. was supported by BBSRC EASTBIO doctoral training partnership (#RG10147-10). This research was supported by grants from NHS Grampian (#RGB4407) and Royal Society (#RG12667-10).

## Author contributions

R.S. supervised the project as the principal supervisor and designed the overall experiments and wrote the manuscript. K.D. also supervised the project as the co-supervisor and designed experiments. H.R. designed and carried out most of the experiments and wrote the manuscript. N.O. analysed the high-throughput RNA-sequencing data provided by Eurofin Genomics. R.E.P.A. helped cell culture, microscopy and flow cytometry. All authors discussed the results and contributed to the manuscript.

## Competing interests

The authors declare no competing interests.
