## [Peer Review File · Communications Biology]

Reviewers' comments:

Reviewer #1 (Remarks to the Author):

The authors developed new protocols for generating TesLCs from ESCs, and show these cells can be used for testicular organoid culturing with EpiLCs to induce PGCLCs. Moreover, TesLC conditioned medium can also induce PGCLCs. The study is well performed and sufficiently interesting to the field. My comments and concerns are:

1. Functional test by transplantation in vivo is preferred.
2. Immunofluorescence should be performed for some key markers, for example, Stra8. Currently only qPCR data.
3. Please check all the scale bars to make sure they are correctly indicated. For example, Extended Figure 1C and 1D.
4. Statistics for P value calculations should be shown either in figure legends or methods.
5. Is there any apoptosis for tc; sf1ESC differentiation in CBF? Why not CBF+IG, since this condition works better for tcESCs.
6. Extended Figure 1C, Day12, the region shown with half area Gata4+, the other half Gata4-, why these cells cluster together? What are these Gata4- cells?
7. Figure 3b, 297T should be 293T

Reviewer #2 (Remarks to the Author):

The Manuscript "Testicular somatic cells derived from embryonic stem cells induce differentiation of epiblasts into germ cells" by Holly Rore et al. describes the development of a protocol for deriving mouse embryonic Sertoli and/or Leydig like cells from undifferentiated mESC and that these cells are able to form testicular organoids supporting germ cell development. The work mostly consists of extensive in vitro differentiation approaches with several genetically modified mESC lines and some data from RNA sequencing analyses.

In vitro modelling of the development of testicular somatic cells has been challenging and the present manuscript provides novel aspects not only on the regulatory pathways and factors in gonadal development but also on in vitro gametogenesis. Indeed, one of the major findings in the study is the capability of the epiblast like cells to integrate with in vitro derived somatic testicular cell like cells and form meiotic germ cells. Besides biological findings, the protocol presented in the manuscript could possibly offer a platform for toxicological studies on environmental toxins and their impact on gonadal development, a topic that is currently highly relevant. The protocol relies on SF1 over-expressing cell line which restricts its' use as a tool both in developmental and toxicological studies but nevertheless, the paper clearly deserves some merit.

While the manuscript is strong enough to be published in Communications Biology, there are few issues that need be addressed.

Major comments

1) The compositions of tested cell culture media and the flow of actual protocols (when and which growth factors or chemicals were added) are difficult to find and follow. The information probably is there but scattered in figures, figure legends and supplemental files. A clear schematic figure (such as Fig 1a) showing all or all the most important conditions (from which results are shown) as their own lines would serve the purpose. Also, the media compositions could be presented as a table in one place. Importantly, the description of the media compositions should be checked as there seem to be conflicting/confusing information. In the main text page 2 line 44 it is stated that the third medium in three step differentiation protocol was RAG supplemented with CHIR, TTNPB and BMP7 while in the extended materials and methods (page 1 line 20) RAG supplementation is mentioned to be only CHIR and BMP7. The extended Figure 1 does not fit either of the descriptions. The actual media compositions are crucial in this kind of paper as the functional protocol is one of the most important finding in the whole study. Clarification in the presentation is

needed.

2) The identity of the CFP positive cells after two step differentiation remains somewhat unverified. Even if Sertoli specific SOX9 is induced, are other crucial factors (such as Sf1, GATA4) also expressed within the same cells? In other words, how Sertoli cell like cells these cells really are? Unfortunately, the "double staining" of SOX9 Sertoli cell specific promoter driven CFP and endogenous SOX9 protein (Fig 1c) does not confirm them to be Sertoli cell like cells but it would be interesting to see how well these two overlapped. Was it 100% or did these two also have separate single staining cell populations? qPCR results show the induction of correct marker genes within the population but the signal may come from distinct cells. Also, the expression of two (Sf1 and Dmrt1) of the five key markers seem to be fairly low at the end of the culture and WT1 does not increase highly at any point of the experiment (Fig 1d), all together NOT supporting the conclusion of the formation of the Sertoli cells. Moreover, the steroidogenic enzyme Hsd3b1 is thought to be foetal Leydig cell marker rather than foetal Sertoli cell marker. Based on the data shown in Fig1 it is fair to say that part of the cells in culture differentiated towards gonadal somatic cells but it does not show or indicate them to be Sertoli like cells more than Leydig like cells. The gonadal cell types present in the culture should be identified in more detail or the text should be revised to indicate the slightly uncharacterized nature of the differentiated cells. The suggestion that constant WNT activation would push the cells towards ovarian direction is reasonable and easy to check by a few qPCR analyses for ovarian markers. The issue of the identity can be seen in the next chapter (data shown in Figure 2) as well. Double staining with GATA4 or some other marker should be shown to confirm the SCLC differentiation. Here also the interpretation of the expression of steroidogenic enzymes is debatable. Did undifferentiated cells at day 0 show significant expression of these enzymes? If not, inductions less than 20 fold from practically zero level are not high or biologically meaningful. Again, the identity of the cells based on steroidogenic enzymes (apart from Hsd17b3) would merely indicate Leydig cell type differentiation.

3) The RNA sequencing analyses are clearly the most underused resources of the present study. Figures 5a and 5b are not explained at all. What do these analyses reveal, show or support? The GO analyses are proper as such and used in here to confirm the identity of the SCLC (something missing in the Figs 1 and 2) but standard differentially expressed (DE) gene analyses could also be presented to show if any novel factors are upregulated during the differentiation. Moreover, the gene lists in extended Figures 5, that could show novel factors, are way too small font for reading. On page 6 line 230 it is stated that the upregulation of Dnmrt1 and FGF9 was evident only after Sf1 induction. As the comparisons are said to be done only with Sf1 induced SCLC, the statement is rather confusing. The data could also be used to show how the in vitro differentiated CFP positive SCLC:s actually differ from their in vivo counterparts giving insights their developmental stage / capacity and possible hints for optimizing the protocol etc. Finally, the chapter is rather challenging to read as the comparisons (somatic and germ cells) are mixed in the text. For clarity, they could be presented as separate studies in their own subchapters.

Minor points that should be clarified

1) In line 56 of the manuscript Cited2 is introduced as steroidogenic marker. Among many other cell types, Cited2 is involved in the development of some steroidogenic cells and even in the regulation of some parts of the steroidogenesis, but yet, it doesn't indicate or mark steroidogenesis.

2) In line 109 it is speculated that due to low DHH secreted from Sertoli cells Leydig cells do not develop and this is why Hsd17b3 and testosterone production remain low. As Hsd17b3 is expressed in Sertoli cells, do you suggest that DHH acts through Leydig cells to induce Sertoli cell Hsd17b3 expression?

DETAILED RESPONSES TO THE REVIEWER'S COMMENTS

We welcome the constructive comments and questions from the reviewers. We have broken up and numbered the comments, so that we could set out our responses in a clear manner. To answer the reviewers' requests, we have reorganised some of the main Figures and added new immunostaining and RT-PCR data, notably in Fig. 1 and 2. We have also removed the image of GATA4 and MVH co-immunostaining because it is less relevant. In addition, we have added more Supplementary Figures and Tables, and uploaded transcriptome data as Supplementary Data in order to respond mainly to the reviewer 2's comments. Some of Supplementary Figures has been relocated, so that figure numbers do not match those in previous version. Furthermore, we have re-worded about some of statements in the main text, particularly toned down the statement about Leydig cell differentiation.

Reviewer #1:

The authors developed new protocols for generating TesLCs from ESCs, and show these cells can be used for testicular organoid culturing with EpiLCs to induce PGCLCs. Moreover, TesLC conditioned medium can also induce PGCLCs. The study is well performed and sufficiently interesting to the field. My comments and concerns are:

1. *Functional test by transplantation in vivo is preferred.*

We are pleased that the reviewer found our work to be valuable and appreciate for the constructive comment. We do plan to transplant TesLCs and tod-PGCLCs into the mouse testis here, but this will be our next goal. We believe the current data have the impacts and the certainty of advance sufficient to interest a broader readership.

2. *Immunofluorescence should be performed for some key markers, for example, Stra8. Currently only qPCR data.*

We have added immunostaining results of OSR1, Sf1, Wt1, and HSD3b in Fig. 1c, 2c and 2g. We have also added Stra8 and SYCP3 results in Supplementary Fig. 5.

3. *Please check all the scale bars to make sure they are correctly indicated. For example, Extended Figure 1C and 1D.*

We have checked them and corrected some of scale bars.

4. *Statistics for P value calculations should be shown either in figure legends or methods.*

The numbers of samples and statistical calculations are added to figure legends.

5. *Is there any apoptosis for tc; sf1ESC differentiation in CBF? Why not CBF+IG, since this condition works better for tcESCs.*

Answer to the first question: Apoptosis does occur during tc;Sf1ESC differentiation. The flow cytometry analysis actually revealed that the percentage of cell death was higher in tc;Sf1ESCs (24.06%) than tcESCs (2.5%) by day 12 (the percentages are not shown in figures). However, we are not sure that these rates simply reflect apoptotic cell death because tc;Sf1ESC-derived TesLCs require stringent trypsinisation due to their firm adhesion to the culture dish. Hence, the cells might have been damaged cells before sorting by flow cytometry.

Answer to the second question: We appreciate that the reviewer understands the reason that we used IGF1 and GDNF. We actually compared CBF with CBF+IG and found that little difference was observed between the two treatments. Therefore, the IG factors were omitted from subsequent cultures. To clarify the point, we have added new statements in page 4 line 121-123.

6. *Extended Figure 1C, Day12, the region shown with half area Gata4+, the other half Gata4-, why these cells cluster together? What are these Gata4- cells?*

As indicated by arrows or arrowheads in Fig. 1e and Supplementary Fig. 1b, TesLCs form a ridge-like structure that has the basement membrane on its edge. So is the case here. To make this point clear, we have added a dotted line demarcating the edge of the ridge-like structure to the image and shown as the Supplementary Fig. 1d. We have also rearranged original Figure 1f and g, and indicated the edge of the ridge by a dotted line in the revised Fig. 1g as well.

We did not characterise the GATA4-negative cells outside the ridge. They could be Leydig cell-like cells, other types of testicular interstitial cell-like cells, or unknown cells. However, we think that they are less important in this image.

7. *Figure 3b, 297T should be 293T*

Amended.

Reviewer #2:

The Manuscript "Testicular somatic cells derived from embryonic stem cells induce differentiation of epiblasts into germ cells" by Holly Rore et al. describes the development of a protocol for deriving mouse embryonic Sertoli and/or Leydig like cells from undifferentiated mESC and that these cells are able to form testicular organoids supporting germ cell development. The work mostly consists of extensive in vitro differentiation approaches with several genetically modified mESC lines and some data from RNA sequencing analyses.

In vitro modelling of the development of testicular somatic cells has been challenging and the present manuscript provides novel aspects not only on the regulatory pathways and factors in gonadal development but also on in vitro gametogenesis. Indeed, one of the major findings in the study is the capability of the epiblast like cells to integrate with in vitro derived somatic testicular cell like cells and form meiotic germ cells. Besides biological findings, the protocol presented in the manuscript could possibly offer a platform for toxicological studies on environmental toxins and their impact on gonadal development, a topic that is currently highly relevant. The protocol relies on SF1 over-expressing cell line which restricts its' use as a tool both in developmental and toxicological studies but nevertheless, the paper clearly deserves some merit.

While the manuscript is strong enough to be published in Communications Biology, there are few issues that need be addressed.

Major comments

1. *The compositions of tested cell culture media and the flow of actual protocols (when and which growth factors or chemicals were added) are difficult to find and follow. The information probably is there but scattered in figures, figure legends and supplemental files. A clear schematic figure (such as Fig 1a) showing all or all the most important conditions (from which*

results are shown) as their own lines would serve the purpose. Also, the media compositions could be presented as a table in one place. Importantly, the description of the media compositions should be checked as there seem to be conflicting/confusing information. In the main text page 2 line 44 it is stated that the third medium in three step differentiation protocol was RAG supplemented with CHIR, TTNPB and BMP7 while in the extended materials and methods (page 1 line 20) RAG supplementation is mentioned to be only CHIR and BMP7. The extended Figure 1 does not fit either of the descriptions. The actual media compositions are crucial in this kind of paper as the functional protocol is one of the most important finding in the whole study. Clarification in the presentation is needed.

We apologise for the confusing statement. We have now improved the schematic diagrams shown in Fig.1a and Supplementary Fig. 1a. We have also replaced Supplementary Fig. 2a with a new table and created a summary table showing all the media components as Supplementary Table 2.

2. The identity of the CFP positive cells after two step differentiation remains somewhat unverified. Even if Sertoli specific SOX9 is induced, are other crucial factors (such as Sf1, GATA4) also expressed within the same cells? In other words, how Sertoli cell like cells these calls really are? Unfortunately, the “double staining” of SOX9 Sertoli cell specific promoter driven CFP and endogenous SOX9 protein (Fig 1c) does not confirm them to be Sertoli cell like cells but it would be interesting to see how well these two overlapped. Was it 100% or did these two also have separate single staining cell populations? qPCR results show the induction of correct marker genes within the population but the signal may come from distinct cells.

The reviewer is correct. In Fig. 1d, as well as Fig 2e, 2f and 2h, the qPCR results do not represent homogeneous SCLC population, but heterogeneous whole TesLC population. That is why we show Supplementary Fig. 3b and 3c, in which qPCR is performed using RNA extracted from sorted CFP-positive and negative cells. The results show enriched expression of SOX9, Gata4 and Sf1 in CFP-positive cells. *Wt1* expression is comparable between CFP-positive and negative cells, which can be attributed to the possibility that the differentiation of renal cells also takes place. We have also added two graphs for the expression of renal markers, *Hoxd11* and *Foxd1* to Fig. 1d.

For further characterisation of TesLCs, we have now added OSR1 and Sf1 immunostaining to Fig. 1c. The presence of OSR1-positive cells ensures that IM-like cells appear by day 5 in the two-step induction process (Fig. 1c). All the Sf1-positive cells express Sox9 by day 8 of culture, although a number of Sox9-single positive cells are also observed. This is presumably because Sf1 expression declines after day 6 as demonstrated by qRT-PCR in Fig. 1d.

3. Also, the expression of two (*Sf1* and *Dmrt1*) of the five key markers seem to be fairly low at the end of the culture. *WT1* does not increase highly at any point of the experiment (Fig 1d), all together NOT supporting the conclusion of the formation of the Sertoli cells.

We acknowledge low expression levels of *Sf1* and *Dmrt1* during tcTesLC differentiation. However we do not agree with the reviewer's comment that our data do not support the formation of Sertoli cell-like cells. As mentioned above, qPCR results indicate relative expression in a heterogeneous population, but do not mean that no Sertoli cell-like cells are present in the population. Regarding *Dmrt1*, it is been shown that *Dmrt1* highly expressed in ESCs. Hence we presume that *Dmrt1* expression progressively decline unless efficient Sertoli cell differentiation is induced. Such is the case shown in Fig. 1d. But, again, this does not mean that Sertoli cell differentiation does not occur.

Wt1 activation apparently looks very low in Fig 1d, but this is an impression caused by scaling the graph in comparison with the other three factors. In fact, actual values are significantly as

high as 19.4, 15.6, 17.0, and 46.4 fold for D6, D8, D10 and D12, respectively. To avoid the visual impression, we have now separated Wt1 graph from the others in Fig. 1d.

4. Moreover, the steroidogenic enzyme Hsd3b1 is thought to be foetal Leydig cell marker rather than foetal Sertoli cell marker. Based on the data shown in Fig.1 it is fair to say that part of the cells in culture differentiated towards gonadal somatic cells but it does not show or indicate them to be Sertoli like cells more than Leydig like cells. The gonadal cell types present in the culture should be identified in more detail or the text should be revised to indicate the slightly uncharacterized nature of the differentiated cells.

We appreciate for the comments that give us the opportunity to reconsider Leydig cell differentiation. We have now added HSD3b and Gata4 double-immunostaining data to Figure 1C and Figure 2e. Intriguingly, almost all HSD3b-positive cells are Gata4-positive even though Gata4 single-positive cells are also present. Gata4 single-positive cells may become SCLCs. The double-positive cells possibly have characteristics of Sertoli cells and Leydig cells since both cells are thought to be derived from multipotent gonadal precursor cells (Svingen and Koopman. Genes Dev 27, 2409-2426. 2013). We have referred to the possibility in the text (in page 5, line 133-135). Because of the co-immunostaining result, we cannot define the Hsd3b1-positive cells as Leydig cell-like cells. Therefore, we have decided to tone down the phrase from 'Leydig cell-like cells' to 'potential steroidogenic cells' (in page 3, line 79). We are of course interested in the potential steroidogenic cells and other uncharacterised cell types. However, it is not the first priority because the present studies aim at showing the existence of Sertoli cell-like cells.

5. The suggestion that constant WNT activation would push the cells towards ovarian direction is reasonable and easy to check by a few qPCR analyses for ovarian markers.

We actually show mean reads of granulosa cell markers in Supplementary Table 1b, which can be sufficient to answer this request. It would be unreasonable for us to test ovarian markers by qPCR in the current paper because we are focusing on the generation of SCLCs from XY ESCs. Even if XY ESCs were prone to differentiate into ovarian follicle cells in the presence of CHIR99021, it would be less important in this work. We actually cultured XX ESCs in the same condition and observed a fewer CFP-positive appeared (unpublished data), suggesting that XX ESCs are preferentially pushed towards ovarian direction. Again, however, this is not relevant to the current paper, but of course we would like to explore it in the future.

6. The issue of the identity can be seen in the next chapter (data shown in Figure 2) as well. Double staining with GATA4 or some other marker should be shown to confirm the SCLC differentiation.

We have now added Sf1 and Wt1 immunohistochemistry data with Sox9 co-staining to Fig. 2c. Almost all *tc;Sf1*TesLCs are consistently Sf1-positive because of forced expression from the Sf1 transgene driven by a ubiquitous promoter. Only a subpopulation of the Sf1-positive cells expresses Sox9, suggesting that they are able to differentiate into other types of testicular somatic cells including steroidogenic cells. On the other hand, the cells positive for Wt1 are identical to Sox9-positive cells. The double-positive cells can be the SCLCs.

7. Here also the interpretation of the expression of steroidogenic enzymes is debatable. Did undifferentiated cells at day 0 show significant expression of these enzymes? If not, inductions less than 20 fold from practically zero level are not high or biologically meaningful. Again, the identity of the cells based on steroidogenic enzymes (apart from Hsd17b3) would merely indicate Leydig cell type differentiation.

We have now added Supplementary Data 1 showing all differentially expressed genes (filtered p adj <0.05, LFC >1 or <-1) between ESCs and SCLCs or E12.5 Sertoli cells. The data show

that there are significant base reads of some steroidogenic genes in ESCs, i.e. at day 0 induction.

Even though the reviewer points out that less than 20-fold activation of Hsd17b3 or other steroidogenic genes might not be good enough to support Leydig cell differentiation, we believe that a considerable amount of testosterone secreted in the medium is a strong evidence for the presence of Leydig cell-like cells in *tc;Sf1*TesLC population. Furthermore, the production of testosterone cannot be necessarily dependent on Hsd17b3, as recent studies revealed that there is a novel testosterone-production pathway independent to *hsd17b3* (Rebourcet et al. FASEB, J. doi.org/10.1096/fj.202000361R. 2020), which has been added to references. However, to satisfy the reviewer, we have again toned down our statement about Leydig cell differentiation and have changed 'Leydig cell-like cells' to 'testosterone-producing cells' in page 6, line 165.

8. Sf1 overexpression on its own may not be sufficient to induction the differentiation of ESCs into Leydig cells.

Previous studies demonstrated that bromoadenosine-cAMP treatment is required for the differentiation of *Sf1*-overexpressing ESCs into Leydig cell-like cells (Jadhav and Jameson, *Endocrinology* 152, 2870-2882, 2011). We have now added the evidence in page 10, line 312 – 314.

9. 3) The RNA sequencing analyses are clearly the most underused resources of the present study. Figures 5a and 5b are not explained at all. What do these analyses reveal, show or support?

We have explained the results in page 8, line 264 and page 9, line 294.

10. The GO analyses are proper as such and used in here to confirm the identity of the SCLC (something missing in the Figs 1 and 2) but standard differentially expressed (DE) gene analyses could also be presented to show if any novel factors are upregulated during the differentiation.

We have added GO enrichment analysis in Supplementary Data 2. We have also explained the results in page 9, line 278-291.

11. Moreover, the gene lists in extended Figures 5, that could show novel factors, are way too small font for reading.

We have improved the image quality. The data has been moved to Supplementary Data. Fig. 6a and b.

12. On page 6 line 230 it is stated that the upregulation of Dnmrt1 and FGF9 was evident only after Sf1 induction. As the comparisons are said to be done only with Sf1 induced SCLC, the statement is rather confusing.

We apologise for the confusing statement. For RNA-seq, we sorted CFP-positive cells from only *tc;Sf1*TesLCs, but not *tc*TesLCs. In other words, CFP-positive cells sorted from *tc;Sf1*TesLCs were used as representative SCLCs, which has been stated in page 8, line 257-258. We then observed significant upregulation of *Dmrt1* and *Fgf9* when SCLC profile were compared to *tc;sf1*ESC profile, but not *tc*ESC profile. However, the result does not mean that the two genes are not upregulated without SF1.

13. *The data could also be used to show how the in vitro differentiated CFP positive SCLC:s actually differ from their in vivo counterparts giving insights their developmental stage / capacity and possible hints for optimizing the protocol etc.*

We appreciate for the comments. We have compared expression profiles of *tc;Sf1*SCLCs and E12.5 Sertoli cells (i.e. *in vivo* counterparts) in this study. However, the comparison is not sufficient to either conclude what stage *tc;Sf1*SCLCs are in Sertoli cell differentiation or give us clues for improving the current protocol. We plan to perform RNA-seq analysis for *tc*SCLCs and also compare expression profiles of *tc*SCLCs and *tc;Sf1*SCLCs with those derived from different developmental stages of *in vivo* Sertoli cells.

14. *Finally, the chapter is rather challenging to read as the comparisons (somatic and germ cells) are mixed in the text. For clarity, they could be presented as separate studies in their own subchapters.*

We have now separated the two studies by creating new paragraphs.

Minor points that should be clarified

15. 1) *In line 56 of the manuscript Cited2 is introduced as steroidogenic marker. Among many other cell types, Cited2 is involved in the development of some steroidogenic cells and even in the regulation of some parts of the steroidogenesis, but yet, it doesn't indicate or mark steroidogenesis.*

We are aware that *Cited2* may not be a definite steroidogenic marker since it was demonstrated that *Cited2* is involve in sex determination, especially transcriptional regulation of *Sry* by genetically interacting with *Sf1* and *Wt1* (Buaas, FW. et al. Hum Mol Genet 18, 2989-3001, 2009). Its expression begins in the adreno-gonadal primordium and is maintained in the adrenal gland, whereas it declines in the gonad after sex determination. To make it clear, we have now referred to *Cited2* as a adrenogonadal marker (in page3 line 71). We have also separated the *Cited2* graph from the others in Supplementary Fig. 1f.

16. 2) *In line 109 it is speculated that due to low DHH secreted from Sertoli cells Leydig cells do not develop and this is why Hsd17b3 and testosterone production remain low. As Hsd17b3 is expressed in Sertoli cells, do you suggest that DHH acts through Leydig cells to induce Sertoli cell Hsd17b3 expression?*

The reviewer is confused. Low Dhh expression may decrease the efficiency of fetal Leydig cell specification. But we do not claim that this is the cause of low Hsd17b3 expression and inefficient testosterone production in fetal Leydig cells, because Hsd17b3 is supposed to be expressed in fetal Sertoli cells. DHH treatment increases testosterone production in whole TesLC population, but we don't know which types of cells produce testosterone in TesLCs. It would be interesting to know whether DHH acts through Leydig cell-like cells to induce Hsd17b3 expression in Sertoli cell-like cells. However, we have no evidence to support the idea right now.

REVIEWERS' COMMENTS:

Reviewer #1 (Remarks to the Author):

The majority of my comments are addressed. A few comments:

1. "Testicular somatic cells" in the title is better to be "Testicular somatic cell-like cells".
2. RNA-Seq data for samples in Figure 5a should be presented in one table (with expression values and genes shown). Same for Figure 5b. This way, the readers can easily compare the value of specific genes in different samples.
3. In Figure 5d, Fgf9 does not seem upregulated in SCLCs when compared to tc;sf1ESC.

Reviewer #2 (Remarks to the Author):

In the revised version of the manuscript most of the questions raised have been addressed sufficiently. Below are replies for a couple of remaining issue:

Major point 3, about Dmrt1 as supporting evidence:

Authors in rebuttal: We acknowledge low expression levels of Sf1 and Dmrt1 during tcTesLC differentiation. However we do not agree with the reviewer's comment that our data do not support the formation of Sertoli cell-like cells. As mentioned above, qPCR results indicate relative expression in a heterogeneous population, but do not mean that no Sertoli cell-like cells are present in the population. Regarding Dmrt1, it is been shown that Dmrt1 highly expressed in ESCs. Hence we presume that Dmrt1 expression progressively decline unless efficient Sertoli cell differentiation is induced. Such is the case shown in Fig. 1d. But, again, this does not mean that Sertoli cell differentiation does not occur.

Reply to authors: My concern was and still is the declining expression of the markers that are supposed to show that differentiation occurs. In the case of Dmrt1 the expression levels at day 12 are clearly (and significantly) lower than in the beginning of the experiment (Fig 1d). The rationale "without differentiation, levels would be even lower" is not very convincing. Please note that this doesn't mean that the differentiation wouldn't occur, other markers, stainings and RNAseq data shows that now, but it means that the disappearing marker in this context is simply not supporting it.

Major points 9-13, about the RNAseq data: The RNAseq data now supports the study clearly better and especially comparisons with E12.5SCs helps to place the cells in the right position when thinking their identity and observed biological properties. Supplemental tables are sufficient and will provide other scientists plenty of novel information to go through. However, a couple of points in the interpretation raises questions. In the discussion (page 10, line 325-327) it is stated "PCA analysis displays less similarities between 326 SCLCs and E12.5SCs than between tod-PGCLCs and E12.5PGCLCs, which may be attributed to heterogeneity of SCLCs even in the sorted cell population." Why do you think they should show similar amount of similarities?

In the next sentence it is concluded that SF1 over-expression doesn't change the nature of the undifferentiated ESCs as there was only 1800 DEGs and just 9 of them were linked to pluripotency. 1800 DEGs is actually a lot and 9 pluripotency related may indeed be highly relevant. Whatever biological differences these 1800 DEGs cause may not be relevant to this study, the outcome with SF1 over-expression is clear. However, the phrase "The results suggest that the two ESCs are essentially similar with respect to pluripotency" is a bit of an overstatement, especially if pluripotency is not tested.

A new comment to revised manuscript: In the page 8 line 246 it is stated that enhanced BMP4 expression would have elicited PGCLC inducing activity (Figure 4e). Did you actually test the effect of BMP4? LDN-inhibition blocks a variety of ALK receptors and hence, multiple TGFb-family members that may be expressed by the cultured cells.

COMMSBIO-20-1870B

RESPONSES TO THE REVIEWER'S COMMENTS

Thank you for the opportunity to finalise our manuscript. We have dealt with the reviewers' concerns as much as possible. We hope our work will be accepted for publication in Communications Biology.

Please find our response to the reviewers' comments (*in italics*) as follows:

Reviewer #1:

The majority of my comments are addressed. A few comments:

1. *"Testicular somatic cells" in the title is better to be "Testicular somatic cell-like cells".*

We agree and have corrected the title.

2. *RNA-Seq data for samples in Figure 5a should be presented in one table (with expression values and genes shown). Same for Figure 5b. This way, the readers can easily compare the value of specific genes in different samples.*

The heatmap of Euclidian distances is a common way to represent overall variation between samples, but not single genes. It would be difficult to present the results in a table. We have already added tables presenting differentially expressed genes with adjusted p-values and log₂ fold changes in Supplementary Data 1, which will help to We would therefore like to leave the Fig 5a and 5b as they are.

3. *In Figure 5d, Fgf9 does not seem upregulated in SCLCs when compared to tc;sf1ESC.*

It can be visual effect cause by colour codes. The actual log₂FC value is indicated in Supplementary Table 1b and Supplementary Data 1c.

Reviewer #2:

In the revised version of the manuscript most of the questions raised have been addressed sufficiently. Below are replies for a couple of remaining issue:

1. *Major point 3, about Dmrt1 as supporting evidence:*

Authors in rebuttal: We acknowledge low expression levels of Sf1 and Dmrt1 during tcTesLC differentiation. However we do not agree with the reviewer's comment that our data do not support the formation of Sertoli cell-like cells. As mentioned above, qPCR results indicate relative expression in a heterogeneous population, but do not mean that no Sertoli cell-like cells are present in the population. Regarding Dmrt1, it is been shown that Dmrt1 highly expressed in ESCs. Hence we presume that Dmrt1 expression progressively decline unless efficient Sertoli cell differentiation is induced. Such is the case shown in Fig. 1d. But, again, this does not mean that Sertoli cell differentiation does not occur.

Reply to authors: My concern was and still is the declining expression of the markers that are supposed to show that differentiation occurs. In the case of Dmrt1 the expression levels at day 12 are clearly (and significantly) lower than in the beginning of the experiment (Fig 1d). The rationale "without differentiation, levels would be even lower" is

not very convincing. Please note that this doesn't mean that the differentiation wouldn't occur, other markers, stainings and RNAseq data shows that now, but it means that the disappearing marker in this context is simply not supporting it.

We appreciate the comment. We have understood what the reviewer meant. Of course, we would like to know the reason that Dmrt1 expression decreases in the future but leave this question unanswered at the moment.

2. *Major points 9-13, about the RNAseq data: The RNAseq data now supports the study clearly better and especially comparisons with E12.5SCs helps to place the cells in the right position when thinking their identity and observed biological properties. Supplemental tables are sufficient and will provide other scientists plenty of novel information to go through. However, a couple of points in the interpretation raises questions.*

In the discussion (page 10, line 325-327) it is stated "PCA analysis displays less similarities between SCLCs and E12.5SCs than between tod-PGCLCs and E12.5PGCLCs, which may be attributed to heterogeneity of SCLCs even in the sorted cell population." Why do you think they should show similar amount of similarities?

We did not mean to show a similar amount of similarities. In fact, we do not expect it since their differentiation processes are independent. Considering the reviewer's comment, we have now changed the statement to "PCA and clustering analyses reveal similarities and/or divergence between samples. The wide divergence between SCLCs and E12.5SCs may be attributed to heterogeneity of SCLCs even after cell sorting because we used only CFP fluorescence for flow cytometry. Use of double or more markers may allow us to sort purer SCLCs".

3. *In the next sentence it is concluded that SF1 over-expression doesn't change the nature of the undifferentiated ESCs as there was only 1800 DEGs and just 9 of them were linked to pluripotency. 1800 DEGs is actually a lot and 9 pluripotency related may indeed be highly relevant. Whatever biological differences these 1800 DEGs cause may not be relevant to this study, the outcome with SF1 over-expression is clear. However, the phrase "The results suggest that the two ESCs are essentially similar with respect to pluripotency" is a bit of an overstatement, especially if pluripotency is not tested.*

We appreciate the comment. We have now deleted the statement "1813 DEGs between *tc*ESCs and *tc*;Sf1ESCs are much fewer than the other comparisons showing several thousands of DEGs, and only 9 of them were categorized into pluripotent stem cell pathway. The results suggest that the two ESCs are essentially similar with respect to pluripotency." because the number of DEGs is less important. We have shown a close similarity between *tc*ESCs and *tc*;Sf1ESCs in Fig. 5c anyway.

4. *A new comment to revised manuscript: In the page 8 line 246 it is stated that enhanced BMP4 expression would have elicited PGCLC inducing activity (Figure 4e). Did you actually test the effect of BMP4? LDN-inhibition blocks a variety of ALK receptors and hence, multiple TGFb-family members that may be expressed by the cultured cells.*

Yes, we did administer BMP4 to EpiLC culture. We tested two conditions; BMP4 only and BMP4 with EGF and SCF (described in Methods). Both conditions gave rise to GFP expression in EpiLCs derived from Prdm1-gfp ESCs. Because the results had been demonstrated by Prof Azim Surani laboratory where Prdm1-gfp ESCs were created (Murakami et al., *Nature* 529, 403-407, 2016), we did not include them in this paper. However, we have now added the evidence and replaced the statement in line 245-249 with "As Murakami et al. previously demonstrated that BMP4 is a key molecule to induce

GFP expression in EpiLCs derived from Prdm1-gfp ESCs (ref. 33), we detected the activation of Bmp4 expression in both CFP-positive and negative populations within tcTesLCs and tc;Sf1TesLCs, although CFP-negative cells showed lower expression in tc;Sf1TesLCs (Fig. 4e)". We also added the reference #33.